# CrossBeam: Learning to Search in Bottom-Up Program Synthesis

**Kensen Shi** [*]
Google Research
kshi@google.com

**Hanjun Dai** [*]
Google Research
hadai@google.com

**Kevin Ellis** [†]
Cornell University
kellis@cornell.edu

**Charles Sutton** [†]
Google Research
charlessutton@google.com

## Abstract

Many approaches to program synthesis perform a search within an enormous space of programs to find one that satisfies a given specification. Prior works have used neural models to guide combinatorial search algorithms, but such approaches still explore a huge portion of the search space and quickly become intractable as the size of the desired program increases. To tame the search space blowup, we propose training a neural model to learn a hands-on search policy for bottom-up synthesis, instead of relying on a combinatorial search algorithm. Our approach, called CrossBeam, uses the neural model to choose how to combine previously-explored programs into new programs, taking into account the search history and partial program executions. Motivated by work in structured prediction on learning to search, CrossBeam is trained on-policy using data extracted from its own bottom-up searches on training tasks. We evaluate CrossBeam in two very different domains, string manipulation and logic programming. We observe that CrossBeam learns to search efficiently, exploring much smaller portions of the program space compared to the state-of-the-art.

## 1 Introduction

Program synthesis is the problem of automatically constructing source code from a specification of what that code should do (Manna & Waldinger, 1971; Gulwani et al., 2017). Program synthesis has been long dogged by the combinatorial search for a program satisfying the specification—while it is easy to write down input-output examples of what a program should do, actually finding such a program requires exploring the exponentially large, discrete space of code. Thus, a natural first instinct is to learn to guide these combinatorial searches, which has been successful for other discrete search spaces such as game trees and integer linear programs (Anthony et al., 2017; Nair et al., 2020).

This work proposes and evaluates a new neural network approach for learning to search for programs, called CrossBeam[1], based on several hypotheses. First, learning to search works best when it exploits the symbolic scaffolding of existing search algorithms already proven useful for the problem domain. For example, AlphaGo exploits Monte Carlo Tree Search (Silver et al., 2016), while NGDS exploits top-down deductive search (Kalyan et al., 2018). We engineer CrossBeam around *bottom-up enumerative search* (Udupa et al., 2013), a backbone of several successful recent program synthesis algorithms (Shi et al., 2020a; Odena et al., 2021; Barke et al., 2020). Bottom-up search is particularly appealing because it captures the intuition that a programmer can write small subprograms first and then combine them to get the desired solution. Essentially, a model can learn to do a soft version of a divide-and-conquer strategy for synthesis. Furthermore, bottom-up search enables execution of subprograms during search, which is much more difficult in a top-down approach where partial programs may have unsynthesized portions that impede execution.

---

[*] Equal contribution.
[†] Equal contribution.
[1] https://github.com/google-research/crossbeam

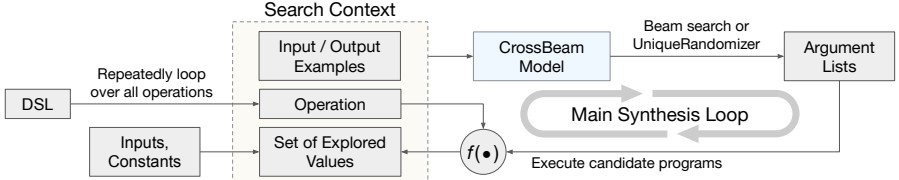

Figure 1: An overview of CROSSBEAM. The search builds a set of explored values, starting with the inputs and constants. It repeatedly loops over all DSL operations, and for each operation, the model takes the search context and produces argument lists for that operation, choosing among the previously-explored values. Executing the operation on the argument lists produces new values.

Second, the learned model should take a "hands-on" role during search, meaning that the learned model should be extensively queried to provide guidance. This allows the search to maximally exploit the learned heuristics, thus reducing the effective branching factor and the exponential blowup. This is in contrast to previous methods that run the model only once per problem (Shi et al., 2020a; Balog et al., 2017), or repeatedly but at lower frequency (Barke et al., 2020).

Third, learning methods should take the global search context into account. When the model is choosing which part of the search space to explore further, its decision should depend not only on recent decisions, but on the full history of what programs have already been explored and their execution results. This is in contrast to hill-climbing or genetic programming approaches that only keep the "best" candidate programs found so far (Schkufza et al., 2013; Shi et al., 2019), or approaches that prune or downweight individual candidate programs without larger search context (Zohar & Wolf, 2018; Odena et al., 2021). Search context can be powerful because one subprogram of the solution may not seem useful initially, but its utility may become more apparent after other useful subprograms are discovered, enabling them to be combined. Additionally, the model can learn context-specific heuristics, for example, to combine smaller expressions for more breadth earlier in the search, and combining larger expressions when the model predicts that it is closer to a solution.

Combining these ideas yields CROSSBEAM, a bottom-up search method for programming by example (Figure 1). At every iteration, the algorithm maintains a search context that contains all of the programs considered so far during search, as well as their execution results. New programs are generated by combining previously-explored programs chosen by a pointer network (Vinyals et al., 2015). The model is trained on-policy using beam-aware training (Negrinho et al., 2018; 2020), i.e., the training algorithm actually runs the CROSSBEAM search, and the loss function encourages the model to progress towards the correct program instead of other candidates proposed by the model. This avoids the potential problems of distribution shift that can arise if the search model is trained off-policy, as in previous works for learning-based synthesis (Devlin et al., 2017; Odena et al., 2021).

On two different domains, we find that CROSSBEAM significantly improves over state-of-the-art methods. In the string manipulation domain, CROSSBEAM solves 62% more tasks within 50K candidate expressions than BUSTLE (Odena et al., 2021) on the same test sets used in the BUSTLE paper. In inductive logic programming, CROSSBEAM achieves nearly 100% success rate on tasks with large enough solutions that prior state-of-the-art has a 0% success rate.

## 2 CROSSBEAM OVERVIEW

In this section we provide an overview of CROSSBEAM, leaving model and training details to Section 3. In our task of program synthesis from input/output examples, we have a domain-specific language (DSL) $\mathcal{L}$ describing a space of programs, and a set of example inputs $\mathcal{I} = \{I_1, \ldots, I_N\}$ and corresponding outputs $\mathcal{O} = \{O_1, \ldots, O_N\}$. The goal is to find a program $P \in \mathcal{L}$ such that $P(I_i) = O_i$ for all $i \in \{1 \ldots N\}$. The DSL $\mathcal{L}$ describes atomic values (constants and input variables) and operations that can be applied to arguments to produce new values. Programs in $\mathcal{L}$ are arbitrarily-nested compositions of operations applied to atomic values or other such compositions.

**Bottom-Up Enumerative Search.** CROSSBEAM extends a basic bottom-up enumerative search algorithm, originally from Udupa et al. (2013) and recently used in other synthesis works (Shi et al., 2020a; Barke et al., 2020; Odena et al., 2021). This basic enumeration considers programs in order of increasing size (number of nodes in the abstract syntax tree), starting from the input variables

---

**Algorithm 1** The CROSSBEAM algorithm, with training data generation in blue

---

**Input:** Input-output examples $(\mathcal{I}, \mathcal{O})$, DSL $\mathcal{L}$ describing constants $Consts$ and operations $Ops$, and (during training) a ground-truth trace $T$

**Output:** A program $P \in \mathcal{L}$ consistent with the examples, and (during training) training data $D_T$

**Auxiliary Data:** A model $M$ trained as described in Section 3 and beam size $K$

  1: $S \leftarrow Consts \cup \mathcal{I}$        ▷ A set of explored values, initially with constants and input variables
  2: $D_T \leftarrow \emptyset$                                      ▷ A set of training datapoints (during training)
  3: **repeat until** search budget exhausted
  4:    **for all** $op \in Ops$ **do**
  5:       $A \leftarrow$ DRAWSAMPLES$(M(S, op, \mathcal{I}, \mathcal{O}), K)$      ▷ Draw $K$ argument lists from the model
  6:       **for all** $[a_1, \ldots, a_n] \in A$ **do**             ▷ $[a_1, \ldots, a_n]$ is an argument list
  7:         $V \leftarrow$ EXECUTE$(op, [a_1, \ldots, a_n])$        ▷ Evaluate the candidate program
  8:         **if** $V \notin S$ **then**           ▷ The value has not been encountered before
  9:            $V.op \leftarrow op, V.arglist \leftarrow [a_1, \ldots, a_n]$     ▷ Store execution history
10:            $S \leftarrow S \cup \{V\}$
11:         **if** $V = \mathcal{O}$ **then**                   ▷ Solution found
12:            **return** $P :=$ EXPRESSION$(V), D_T$
13:      **if** $IsTraining \wedge T[0].op = op$ **then**
14:         $D_T \leftarrow D_T \cup \{((S, op, \mathcal{I}, \mathcal{O}), T[0].arglist)\}$     ▷ Save model inputs and ground-truth
15:         **if** $T[0].arglist \notin A$ **then**       ▷ If beam search did not generate the ground-truth
16:            $S \leftarrow S \cup \{T[0]\}$        ▷ Continue as if we did generate the ground-truth
17:         $T.pop(0)$

---

and DSL constants. For each program, the algorithm stores the value that results from executing the program on the inputs. Then, at each iteration of search, it enumerates all programs of the given target size, which amounts to enumerating all ways of choosing a DSL operation and choosing its arguments from previously-explored values, such that the resulting program typechecks and has the target size. Each candidate program results in a new value, which is added to the set of explored values if it is not semantically equivalent (with respect to the I/O examples) to an existing value. Values constructed in this way store the operation and argument list used. Once we encounter a value that is semantically equivalent to the example outputs, we have found a solution and can recursively reconstruct its code representation. This search algorithm is complete, but the search space grows exponentially so it quickly becomes intractable as the size of the solution program grows.

**CROSSBEAM Algorithm.** To combat the exponential blowup of the search space, the CROSSBEAM approach (Figure 1, Algorithm 1) uses a neural model in place of complete enumeration in the basic bottom-up search. We still build a set of explored values, but we no longer enumerate all argument lists nor consider programs strictly in order of expression size. Instead, the model determines which programs to explore next, that is, which previously-explored values to combine in order to generate the next candidate program. More specifically, we repeatedly loop over all operations in the DSL. For each operation, the model takes as input the entire search context, including the I/O examples, the current operation, and the set $S$ of previously-explored values. The model defines a distribution over argument lists for the operation, in the form of pointers to existing values in $S$. From this distribution, we can draw argument lists from the model using sampling or beam search, to produce candidate programs. Finally, we execute each of the candidate programs and add the resulting values to $S$, with pruning of semantically-equivalent values as in the basic bottom-up search. The search continues until a solution is found or the search budget is exhausted.

**UniqueRandomizer.** We use beam search to draw argument lists from the model during training. Because the model may choose the same argument list for the same operation in different iterations of search, the search algorithm may stall if no new values are explored. This is especially problematic during evaluation, where we may search for a long time. Therefore, during evaluation, instead of beam search we use UniqueRandomizer (Shi et al., 2020b), which draws distinct samples from a sequence model incrementally by storing the previous samples in a trie so as to avoid duplicates. Using UniqueRandomizer, we draw distinct samples (argument lists) until we produce $K$ values that have not been seen yet, up to a budget of $\lambda \cdot K$ samples per operation per iteration, where $K$ is the "beam size" and $\lambda$ is a hyperparameter describing how hard we should try to produce $K$ new values. During evaluation in our experiments, we set $K = \lambda = 10$.

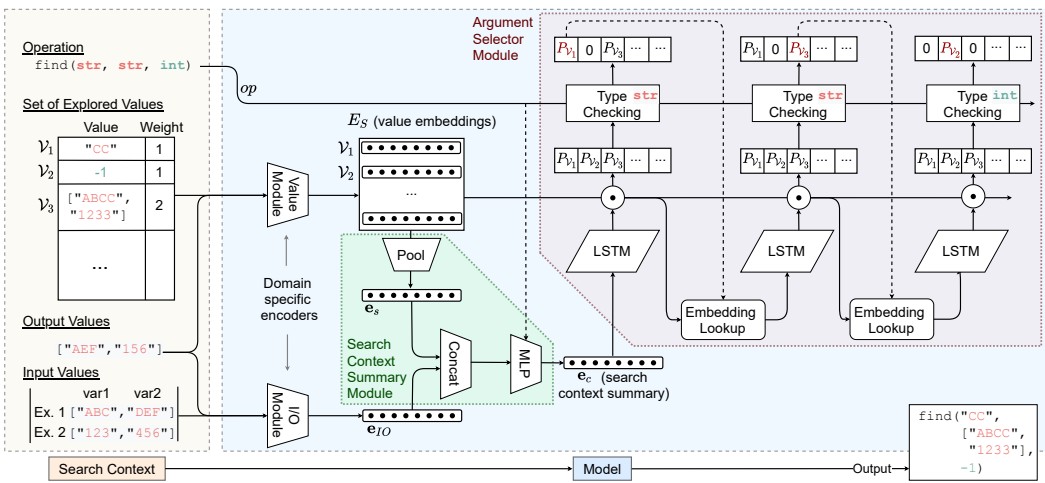

Figure 2: Model that proposes new candidate programs during search. Each new candidate program consists of an operation and an argument list. Each argument in the list is a pointer to a value produced from a previously explored program.

## 3 MODEL DETAILS

Our neural model for CROSSBEAM is a policy for bottom-up search. It takes as input an operation in the DSL and a search context, and outputs a distribution over argument lists for the operation, in the form of pointers to previously-explored values. The model has four components (Figure 2): (1) the I/O module, (2) the value module, (3) the search context summary module and (4) the argument selector module. For different domains one can make different design choices for these modules. For example, in the string manipulation domain, we leverage property signatures (Odena & Sutton, 2020) in the same way as in BUSTLE (Odena et al., 2021), and for the logic programming domain, we propose two variants based on the MLP or GREAT Transformer (Hellendoorn et al., 2019) architectures. Below we present the generic design of these modules; for details, see Appendix A.

**I/O module.** The I/O module takes the input-output examples $(\mathcal{I}, \mathcal{O})$ as input and produces a $d$-dimensional vector representation $\boldsymbol{e}_{IO} \in \mathbb{R}^d$ that summarizes the specification. Depending on the application domain, one can use RNNs (Devlin et al., 2017), property signatures (Odena & Sutton, 2020), or other domain-specific approaches to embed the I/O example into $\boldsymbol{e}_{IO}$.

**Value module.** The value module embeds the set of explored values $S$ into a matrix $E_S \in \mathbb{R}^{|S| \times d}$. The $i$-th row of the matrix, $E_{S_i}$, is the embedding of the $i$-th explored value $V_i \in S$. Each value embedding $E_{S_i} = \boldsymbol{s}_i + \boldsymbol{z}_i$ sums up two components. First, $\boldsymbol{s}_i$ is a domain-specific embedding that can be implemented differently for different domains, but likely using similar techniques as the I/O module. Second, a size embedding $\boldsymbol{z}_i$ makes the search procedure aware of the "cost" of the value $V_i$. To implement this, we compute $\min\{size(V_i), m_s\}$ and look it up in an embedding matrix in $\mathbb{R}^{m_s \times d}$ to get the size embedding $\boldsymbol{z}_i$, where $m_s$ is a maximum size cutoff for embedding purposes.

**Search context summary module.** This module summarizes the search context, which includes the set of explored values $S$, the I/O examples $(\mathcal{I}, \mathcal{O})$, and the current operation $op$, into a vector representation $\boldsymbol{e}_c \in \mathbb{R}^d$. The search context summary $\boldsymbol{e}_c$ would then be given to the argument selector module. Since the cardinality of $S$ changes during the search procedure, and it should be permutation-invariant by nature, we can use an approach like DeepSets (Zaheer et al., 2017) to get a summary $\boldsymbol{e}_s$ for $E_S$. In particular, we set $\boldsymbol{e}_s$ to be the max-pooling over $E_S$ across the values. We then obtain $\boldsymbol{e}_c = \mathrm{MLP}_{op}([\boldsymbol{e}_s, \boldsymbol{e}_{IO}])$ via an operation-specific MLP on top of the concatenation of the value embedding summary $\boldsymbol{e}_s$ and the I/O embedding $\boldsymbol{e}_{IO}$.

**Argument selector module.** With all of the context provided above, the argument selector module is designed to pick the most likely sequences of arguments in the combinatorially large space of size $|S|^{op.arity}$, where $op.arity$ is the arity of the currently chosen operation. Since the arity of each op-

eration is typically small, we can use an autoregressive model (in our case an autoregressive LSTM) to approximately output the argument list with maximum likelihood with $O(|S| \times op.arity)$ cost. This also allows us to perform beam search to approximately select the $K$ most likely combinations of arguments. Our LSTM uses the search context summary $e_c$ as the initial hidden state. At each argument selection step $t \in \{1, \ldots, op.arity\}$ we use the LSTM's output gate $h_t \in \mathbb{R}^d$ to select the argument from $S$. Unlike in the language modeling case where the decoding space is given by a fixed vocabulary, here we have a growing vocabulary $S$. Inspired by pointer networks (Vinyals et al., 2015), we model the output distribution at the $t$-th step as $p(V_i \mid h_t, S) = \frac{\exp(h_t^\top E_{S_i})}{\sum_{j=1}^{|S|} \exp(h_t^\top E_{S_j})}$. Since not all values are type compatible for the $t$-th argument of operation $op$, we mask out infeasible values before computing $p(V_i \mid h_t, S)$. After the model selects a certain argument $a_t \in S$ at step $t$, the LSTM updates the hidden states with the embedding $E_{S_{a_t}}$ and proceeds with the next step if $t < op.arity$, or stops if we have obtained an entire argument list for $op$.

**Training.** Unlike typical supervised learning, our model uses context that results from searching according to the model. One approach is to provide supervision for how the model should proceed given a randomly-generated search context. However, the resulting distribution shift could lead to inferior generalization. Thus, we instead train the model on-policy, where we run the model in a search to produce realistic search contexts for training. Because the training data includes the model's previous decisions in the search context, the model learns to correctly continue the search, e.g., extending promising search directions or recovering from mistakes.

The overall training algorithm can be found in Algorithm 1. The main idea is that we run search during training and collect supervised training examples for the model as it searches. Given a task specified by $(\mathcal{I}, \mathcal{O})$ with a ground-truth program $P$, we create a trace $T$, which is a list of steps the search should take to construct $P$ in a bottom-up fashion. Each element $T[i] \in T$ contains an operation $T[i].op$ and an argument list $T[i].arglist$ that builds one node of the abstract syntax tree for $P$. Then, in each search step with context $S$ where we are considering the operation $T[i].op$, we collect a training example with input $(S, T[i].op, \mathcal{I}, \mathcal{O})$ and target $T[i].arglist$. Note that we do not make a separate copy of $S$ for each training example, but rather we record the current size of $S$ and refer to the final $S$ for each task, since $|S|$ grows monotonically during the search. During the search we also add $T[i]$ to $S$, so that we may construct correct argument lists involving $T[i]$ in future steps. We use collected training examples to train the model using standard maximum likelihood training.

Note that $T$ is not necessarily unique for a given program $P$, because the syntax tree for $P$ could be linearized in different orders. In this case, we sample a ground-truth trace randomly from the possible orderings. We denote $T \equiv P$ if $T$ is a possible trace of $P$. We then formulate the training loss as the expectation of the negative log-likelihood of the ground-truth argument lists under the corresponding search context and current model parameters $\theta$:

$$\mathcal{L}(P, \mathcal{I}, \mathcal{O}; \theta) = \mathbb{E}_{T:T \equiv P} \left[ \frac{1}{|D_T|} \sum_{((S, op, \mathcal{I}, \mathcal{O}), arglist) \in D_T} - \log p(arglist \mid S, op, \mathcal{I}, \mathcal{O}; \theta) \right] \quad (1)$$

We take gradient descent steps in between searches on the training tasks, so that the model improves over time while producing better-quality data for gradient descent. To generate training tasks, we use a technique from prior work (Shi et al., 2020a; Odena et al., 2021) where we run the bottom-up enumerative search starting from different random inputs, randomly selecting explored values to serve as target outputs. Even though CROSSBEAM is trained using randomly-generated tasks, our experiments show that it still performs well on realistic test tasks. We train CROSSBEAM with a distributed synchronized Adam optimizer on 8 V100 GPUs, with gradient accumulation of 4 for an effective batch size of 32. We generate 10M training tasks for string manipulation and 1M for logic programming. The models train for at most 500K steps or 5 days, whichever limit is reached first. We use $K = 10$ and learning rate $1 \times 10^{-4}$ during training. After every 10K training steps, we evaluate on synthetic validation tasks to choose the best model.

## 4 EXPERIMENTS

Our experiments compare CROSSBEAM to state-of-the-art methods in two very different domains, string manipulation and inductive logic programming.

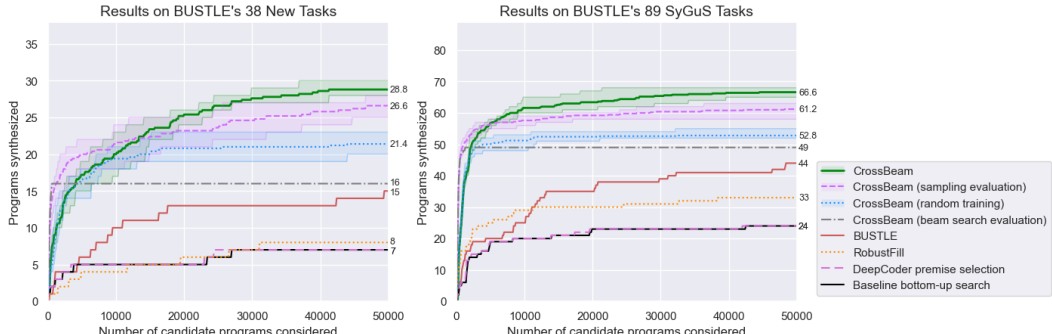

Figure 3: Results on the two sets of benchmark tasks used in the BUSTLE paper. For non-deterministic CROSSBEAM variations, we plot the mean over 5 trials with shading between the minimum and maximum results over those 5 trials. CROSSBEAM is $15.9\times$ (left) or $25.5\times$ (right) more efficient than BUSTLE, in terms of the number of candidate programs needed to reach BUSTLE's 15 solves (left) or 44 solves (right). Overall, CROSSBEAM solves 62% more tasks than BUSTLE.

## 4.1 STRING MANIPULATION

For the string manipulation domain, we use the same DSL, success criteria (satisfying the I/O examples), training task generation procedure, and test tasks as used in BUSTLE (Odena et al., 2021). See Appendix B for details. The BUSTLE approach extends the same enumerative bottom-up search as CROSSBEAM, but BUSTLE trains a neural model to predict how useful values are (i.e., whether a value is a subexpression of a solution) and prioritizes values during the search accordingly. BUSTLE is still a complete enumerative search and will *eventually* explore the entire space of programs in the DSL, and thus it still suffers from the issue of search space blowup, albeit to a much lesser degree than plain enumeration. In contrast, CROSSBEAM trades off completeness for exploration efficiency, as we have no guarantee that the model will eventually explore the entire search space.

We run CROSSBEAM on both sets of test tasks used to evaluate BUSTLE, with a search budget of 50,000 candidate programs. Every value obtained by applying an operation to an argument list, including those pruned due to semantic equivalence, is a "candidate program." Figure 3 shows the number of programs synthesized as a function of the number of candidate programs considered. Because CROSSBEAM is not deterministic due to sampling argument lists with UniqueRandomizer, we run it 5 times and plot the mean, minimum, and maximum performance.

We borrow the following comparisons and results from the BUSTLE paper. (1) BUSTLE refers to the best approach in the paper, i.e., using both the model and heuristics. (2) The baseline bottom-up enumerative search is the same as described in Section 2. (3) RobustFill (Devlin et al., 2017) is an LSTM that predicts end-to-end from the I/O examples to the program tokens. For RobustFill, a "candidate program" is one unique sample from the generative model, obtained via beam search and ordered by decreasing likelihood. (4) DeepCoder-style (Balog et al., 2017) premise selection uses a neural model to predict which DSL operations will be used given the I/O examples. The least likely operations according to the model are removed, and then the baseline bottom-up search is run.

Figure 3 shows that CROSSBEAM significantly outperforms the other methods on both sets of benchmarks, on average solving 36.4 more problems in total compared to BUSTLE (an improvement of 62%). In fact, in order to match BUSTLE's performance after 50,000 candidate programs, CROSSBEAM considers about $20\times$ fewer candidate programs. CROSSBEAM still outperforms BUSTLE when using a much larger budget of 1 million candidate programs (Appendix C). Thus, we conclude that CROSSBEAM explores the search space much more efficiently than the prior approaches.

**Ablations.** We perform an ablation study to quantify the effects of training the model on-policy and using UniqueRandomizer during evaluation. We try the following 3 variations of CROSSBEAM:

1. Random training: instead of using beam search on the model to produce argument lists $A$ during training, we instead obtain argument lists by randomly sampling values from the set $S$ of explored values. As the search progresses, the search context (specifically the set $S$) diverges from what the model would produce during evaluation. Hence, this variation trains the model off-policy.

**A**

| | |
|---|---:|
| $p(X,Y) = \exists Z : q(X,Z) \wedge r(Z,Y)$ | JOIN |
| $p(X,Y) = q(Y,X)$ | TRANSPOSE |
| $p(X) = q(X) \vee r(X)$ | $\left.\vphantom{\begin{matrix}a\\b\end{matrix}}\right\}$ DISJUNCT |
| $p(X,Y) = q(X,Y) \vee r(X,Y)$ | |
| $p(X) = b(X) \vee \exists U,V : q(X,U) \wedge r(U,V) \wedge p(V)$ | $\left.\vphantom{\begin{matrix}a\\b\end{matrix}}\right\}$ RECURSE |
| $p(X,Y) = b(X,Y) \vee \exists U,V : q(X,U) \wedge r(Y,V) \wedge p(U,V)$ | |
| $p(X) = q(X,X)$ | CAST2→1 |
| $p(X,Y) = q(X) \wedge q(Y)$ | CAST1→2 |
| $\text{zero}(X) = (X = 0)$ | (primitive) |
| $\text{succ}(X,Y) = (X + 1 = Y)$ | (primitive) |
| $\text{eq}(X,Y) = (X = Y)$ | (primitive) |

**B**

| Program in DSL | Translation to Prolog |
|---|---|
| RECURSE( | `both_zero(X,Y) :- zero(X), zero(Y).` |
| TRANSPOSE(zero$_{1\rightarrow 2}$), | `sub1(X,Y) :- succ(Y,X).` |
| TRANSPOSE(successor), | `add2(X,Y) :- succ(X,Z), succ(Z,Y).` |
| TRANSPOSE( | `add3(X,Y) :- succ(X,Z), add2(Z,Y).` |
| JOIN(successor, | `sub3(X,Y) :- add3(Y,X).` |
| JOIN(successor, | `p(X,Y) :- both_zero(X,Y).` |
| successor)))) | `p(X,Y) :- sub1(X,U), sub3(Y,V), p(U,V).` |

Figure 4: **A.** (Top) Our four operators for building logic programs. Each operator builds a new predicate $p$ from smaller predicates ($q$, $r$, $b$). The disjunct and recursion operators automatically have different forms depending on the arity of their arguments. (Middle) We assume each predicate has arity 1 or 2, and automatically cast the arity of input predicates as needed using CAST2→1 and CAST1→2. (Bottom) The three primitive relations given to the system. **B.** Example program for the relation $3x = y$. A cast from arity 1 to arity 2 (zero$_{1\rightarrow 2}$) is performed automatically. This size 11 program represents a recursive Prolog routine with 7 clauses and 5 invented predicates, equivalent to saying that the relation $3x = y$ holds for $(0,0)$ and that it holds for $(x,y)$ if it holds for $(x-1, y-3)$.

2. Beam search evaluation: instead of using UniqueRandomizer to draw argument lists $A$ during evaluation, we simply perform beam search with the same beam size $K = 10$. Note that the beam search is deterministic, so the search is prone to stalling if all argument lists result in values that are semantically equivalent to values already explored, leading to repeated looping without progress. This stalling behavior is quite apparent in our experimental results.

3. Sampling evaluation: instead of using UniqueRandomizer to draw argument lists during evaluation, we sample $K = 10$ argument lists according to the distribution given by the model. This leads to more randomness in the argument lists and less stalling, but some stalling can still occur if the high-probability argument lists all produce values semantically equivalent to ones already explored. This motivates our use of UniqueRandomizer in normal CROSSBEAM, where we sample up to $\lambda \cdot K$ argument lists without replacement in search for semantically different values.

As shown in Figure 3, all of these ablations perform worse than normal CROSSBEAM but better than all prior approaches. Interestingly, using beam search or sampling for evaluation results in the method being much more efficient when there are few candidate programs considered (i.e., when the search has not yet stalled), but using UniqueRandomizer allows normal CROSSBEAM to reach higher performance later as the search continues to explore thousands of candidates.

**Wallclock Time.** Despite CROSSBEAM's impressive performance when controlling for the number of expressions considered, CROSSBEAM does not quite beat BUSTLE when we control for wallclock time (but it does outperform the baseline bottom-up search). However, this comparison is not completely fair to CROSSBEAM. We compare to the BUSTLE implementation from the original authors, but this is an optimized Java implementation and was designed to enable easy batching of the model predictions. In contrast, CROSSBEAM is implemented in Python, and batching model predictions for all operations in an iteration and batching the UniqueRandomizer sampling are both feasible but challenging engineering hurdles that would lead to speedups, but are currently not implemented. In Appendix D we discuss this comparison further with quantitative experimental results.

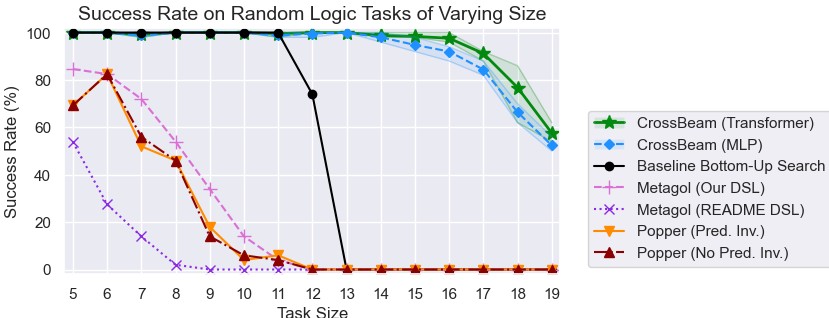

Figure 5: Success rate of different methods on randomly-generated logic tasks in 30 seconds. CROSSBEAM maintains very high success rates on the more difficult tasks where prior works fail.

## 4.2 INDUCTIVE LOGIC PROGRAMMING

Inductive Logic Programming or ILP (Cropper et al., 2021) is the program synthesis problem of learning logical relations (predicates) from examples, which we specify with truth tables. Here, these predicates are represented by logic programs (Prolog or Datalog; see Bratko (2001)) expressing first-order logical statements with quantified variables and recursion. ILP systems have been used to make inferences over large knowledge graphs (Muggleton & Lin, 2013), do programming-by-example (Lin et al., 2014), and perform common sense reasoning (Katz et al., 2008).

Search in ILP is especially difficult when the learned programs can be recursive, and when they can define auxiliary "invented" predicates (analogous to defining helper subroutines). As such, most ILP systems limit themselves to learning a handful of logical clauses, and many forego either recursion or predicate invention. We apply CROSSBEAM to ILP with the goal of synthesizing larger programs with both recursion and predicate invention. We first define a DSL over predicates (Figure 4) that works by building larger predicates out of smaller ones, starting from primitive predicates computing successorship, equality, and the "zero" predicate. Because CROSSBEAM executes every partially-constructed program, we think of each predicate as evaluating to the set of tuples of entities for which the predicate is true. To ensure this set remains small, we bound the arity of synthesized predicates (arity $\leq 2$) and work with entities in a relatively small domain of numbers in $\{0, \ldots, 9\}$.

We study CROSSBEAM's ILP abilities by synthesizing both randomly-generated and handcrafted predicates from examples, with the goal of answering the questions of whether the system can perform predicate invention, whether it can learn recursive programs, and how it compares to state-of-the-art ILP systems. Our synthetic test set contains tasks of varying size, i.e., the minimum number of nodes in the expression tree of a solution. For each size in $\{5, \ldots, 19\}$, we randomly select 50 tasks among all possible tasks of that size, excluding those appearing in the training dataset.[2] Our handcrafted test set, inspired by Peano arithmetic, comprises 30 common mathematical relationships, e.g., whether a number is even, or if a number is greater than another. For each synthesis task, the system is given every positive/negative example for numbers in $\{0, \ldots, 9\}$ as a truth table.

We run CROSSBEAM with either a MLP- or Transformer-based value encoder for predicates (see Appendix A.2). We compare with the classic Metagol (Muggleton & Lin, 2013), a state-of-the-art system Popper (Cropper & Morel, 2021a), and the basic bottom-up enumerative search. We run all methods for a time limit of 30 seconds per task. CROSSBEAM significantly outperforms all of the other methods in both test sets. We summarize our findings here with more details in Appendix E.

Figure 5 shows the results on the randomly-generated test set, where CROSSBEAM achieves close to 100% success rate on tasks up to size 16 and over 50% success rate on tasks up to size 19, while the success rates of all other methods drop to 0% by size 13. This is evidence that CROSS-BEAM is a major forward step in tackling the issue of search space explosion. For the handcrafted tasks, CROSSBEAM achieves a 93% success rate, versus a 60% success rate for Metagol and Popper (Figure 7 in Appendix E). CROSSBEAM also does not appear to struggle with recursion (every hand-crafted task can only be solved with recursion) and consistently solves problems requiring predicate

---

[2]We ran the bottom-up search to exhaustively explore all distinct values up to size 19 inclusive, which takes about 6 hours. We sampled 1 million of those values to serve as training tasks. For sizes 5 and 6, there were fewer than 50 tasks not used in training, so we included all such tasks in our synthetic test set.

invention (Popper without predicate invention solves 37%, which serves as a guide to what fraction of the problems require defining auxiliary predicates).

To our surprise, the baseline bottom-up enumeration also outperforms Metagol and Popper, at least on our test sets where the domain of the predicates is small. To our knowledge, bottom-up enumeration (and pruning based on observational equivalence) has never been tried for ILP before.

## 5 RELATED WORK

Machine learning for program synthesis has been an active area (Gottschlich et al., 2018; Gulwani et al., 2017; Allamanis et al., 2018). For example, DeepCoder (Balog et al., 2017) uses a learned model to select useful operations once at the beginning of search. Although this idea is pragmatic, the disadvantage is that once the search has started, the model can give no further feedback. Odena & Sutton (2020) use property signatures within a DeepCoder-style model for premise selection.

Many learning based approaches to synthesis can be viewed as using learning to guide search. Some methods employ models that emit programs token-by-token (Bunel et al., 2018; Devlin et al., 2017; Parisotto et al., 2017), which can be interpreted as using learning to guide beam search. Recent versions of this idea use large pretrained language models to generate programs (Chen et al., 2021; Austin et al., 2021). Rubin & Berant (2021) builds a semi-autoregressive bottom-up generative model for semantic parsing, where the beam decoding is embedded in the model decoding to achieve logarithmic runtime. Alternately, top-down search can be guided by learning with a syntax guided search over programs (Yin & Neubig, 2017; Lee et al., 2018). Another line of work uses learning to guide a two-level search by first generating a sketch of the program (Nye et al., 2019; Murali et al., 2018) or latent representation (Hong et al., 2021). None of this work uses execution information to guide the search, or uses the rich information from the search context, as CROSSBEAM does.

*Execution-guided neural synthesis* methods use the results of executing partial programs to guide search, which is a powerful source of information. Zohar & Wolf (2018) learns to write straight-line code line-by-line, and to ignore ("garbage collect") lines deemed irrelevant to further search. A similar approach is to rewrite a programming language so that it can be evaluated "left-to-right," allowing values to be used to prioritize a tree search in a reinforcement learning framework (Ellis et al., 2019). Similarly, Chen et al. (2019) use intermediate values while synthesizing a program using a neural encoder-decoder model, but again this work proceeds in a variant of left-to-right search that is modified to handle conditionals and loops. To illustrate the potential differences between left-to-right and bottom-up methods, imagine that the correct program is `f(A,B)`, where `A` and `B` are arbitrary subprograms. Then if a left-to-right search makes a mistake in generating `A`, that search branch will never recover. In contrast, because bottom-up search maintains a population of many partial programs, it can build up subprograms `A` and `B` roughly independently. While all these execution-guided methods have "hands-on" neural models, our approach differs by looking at the global search context—and it learns to do so effectively because it learns on-policy.

Our string domain is inspired by the symbolic synthesizer FlashFill (Gulwani, 2011) and subsequent work on learning-based synthesis for string manipulation (Menon et al., 2013; Odena et al., 2021). Our use of imitation to guide search is inspired by work in learning to search (Daumé III et al., 2009; Ross et al., 2011; Chang et al., 2015) and beam-aware training (Negrinho et al., 2018; 2020).

## 6 CONCLUSION

CROSSBEAM combines several powerful ideas for learning-guided program synthesis. First, bottom-up search provides a symbolic scaffolding that enables execution of subprograms during search, which provides an important source of information for the model. Second, the learned model takes a "hands-on" role during search, having sufficient freedom to manage the effective branching factor of search. Third, the model takes the search context into account, building a continuous representation of the results of all programs generated so far. Finally, training examples are generated on-policy during search, so that the model learns correct followups for its previous decisions. Our experiments show that this is an effective combination: in the string manipulation domain, CROSSBEAM solves 62% more tasks than BUSTLE, and in the logic domain, CROSSBEAM achieves nearly 100% success rate on difficult tasks where prior methods have a 0% success rate.

ACKNOWLEDGMENTS

The authors would like to thank Rishabh Singh and the anonymous reviewers for their helpful comments.

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

# Appendix

## A  DETAILS OF MODEL ARCHITECTURES

### A.1  MODEL DESIGN FOR STRING MANIPULATION

We follow the BUSTLE paper (Odena et al., 2021) in using property signatures (Odena & Sutton, 2020) to featurize individual values and relationships between two values, using properties such as "does the string contain digits," "is the integer negative," and "is the first string a substring of the second." We use the same set of properties as used in BUSTLE. Given a set of properties and one or two values to featurize, we can compute whether that property is always true across examples, always false across examples, sometimes true and sometimes false, or not applicable (due to type mismatch). By evaluating all properties on a value and its relationship to the target output, we obtain a *property signature* used in the I/O module and the value module.

**I/O module.** As in BUSTLE, we use property signatures to featurize all of the input variables and the output, as well as the relationships between each input variable and the output. The result is a feature vector ("property signature") $\boldsymbol{f}_{IO} \in C^{L_{IO}}$ where $C = \{true, false, mixed, N/A\}$ is the set of possible results for a single property, and $L_{IO}$ is the length of the property signature for the entire I/O example. We project each element of $C$ into a 2-dimensional embedding, pass each signature element $f_{IO_i} \in C$ through this embedding, and project the concatenation of the embeddings into $\boldsymbol{e}_{IO}$ using a multilayer perceptron $\text{MLP}_{IO}$.

**Value module.** Here we again follow BUSTLE and compute the property signature $\boldsymbol{f}_{V_i} \in C^{L_{val}}$, where $L_{val}$ is the length of a property signature when we featurize $V$ individually and compared to the output $\mathcal{O}$. As in the I/O module, we pass $\boldsymbol{f}_{V_i}$ through a 2-dimensional embedding followed by a multilayer perceptron $\text{MLP}_{val}$, to obtain the signature embedding $\boldsymbol{s}_i$.

### A.2  MODEL DESIGN FOR LOGIC PROGRAMMING

In the logic programming domain, every program is a predicate with arity 1 or 2 over the domain of entities $\{0, \ldots, 9\}$. We think of each such predicate as computing a binary-valued tensor with shape $(10, )$ for arity-1 predicates, or $(10, 10)$ for arity-2 predicates. Both the I/O specification and program values are modeled as tensors of these shapes. For the I/O module and the value module, we consider two different architectures for encoding these tensors.

**MLP Encoder.** This encoder flattens the binary tensor representing the predicate, pads it with zeros to have length 100 (as binary relations can have size $10 \times 10 = 100$), and prepends a single binary variable indicating the arity of the relation. The resulting 101-dimensional object is processed by a 4-layer MLP with 512 hidden units and ReLU activations.

**Relational Transformer.** This encoder embeds each element of the domain of entities (the numbers $\{0, \ldots, 9\}$) and then applies a relational transformer to these entities. Our relational transformer is based on GREAT (Hellendoorn et al., 2019).[3] Each layer of self-attention is modulated by the truth-value judgment of the predicate we are encoding. Concretely, the unnormalized attention coefficient between entity $i$ and entity $j$ in layer $l$ of the transformer, written $\alpha_{ij}^l$, is a function of the key $k_j^l$, the query $q_i^l$, and the embedding of the relation between those entities, written $r_{ij}^l$:

$$\alpha_{ij}^l = (q_i^l + r_{ij}^l)^\top k_j^l \tag{2}$$

The relation embedding $r_{ij}^l$ is a function of the predicate we are encoding, written $p$. The intuition is that we want different valuations of $p(i)$, $p(j)$, $p(i, j)$, and $p(j, i)$ to be embedded as different vector representations. We also want the relation to be a function of whether we are encoding a value or the specification (written "spec"). We define the following function, which assigns a unique integer from 0 to 15 to every unique relation on $\leq 2$ inputs:

$$\text{UNIQUEID}(p, i, j) = 1 \cdot p(i) + 2 \cdot p(j) + 4 \cdot \mathbb{1}\,[p \text{ has arity } 2]$$
$$+ 1 \cdot p(i, j) + 2 \cdot p(j, i) + 8 \cdot \mathbb{1}\,[p = \text{spec}]$$

---

[3] The model differs from GREAT only in that the relation modulates attention coefficients via a vector rather than a scalar.

$$\text{Expression } E := S \mid I \mid B$$
$$\text{String expression } S := \text{Concat}(S_1, S_2) \mid \text{Left}(S, I) \mid \text{Right}(S, I) \mid \text{Substr}(S, I_1, I_2)$$
$$\mid \text{Replace}(S_1, I_1, I_2, S_2) \mid \text{Trim}(S) \mid \text{Repeat}(S, I) \mid \text{Substitute}(S_1, S_2, S_3)$$
$$\mid \text{Substitute}(S_1, S_2, S_3, I) \mid \text{ToText}(I) \mid \text{LowerCase}(S) \mid \text{UpperCase}(S)$$
$$\mid \text{ProperCase}(S) \mid T \mid X \mid \text{If}(B, S_1, S_2)$$
$$\text{Integer expression } I := I_1 + I_2 \mid I_1 - I_2 \mid \text{Find}(S_1, S_2) \mid \text{Find}(S_1, S_2, I) \mid \text{Len}(S) \mid J$$
$$\text{Boolean expression } B := \text{Equals}(S_1, S_2) \mid \text{GreaterThan}(I_1, I_2) \mid \text{GreaterThanOrEqualTo}(I_1, I_2)$$
$$\text{String constants } T := \text{""} \mid \text{" "} \mid \text{","} \mid \text{"."} \mid \text{"!"} \mid \text{"?"} \mid \text{"("} \mid \text{")"} \mid \text{"["} \mid \text{"]"} \mid \text{"<"} \mid \text{">"}$$
$$\mid \text{"\{"} \mid \text{"\}"} \mid \text{"-"} \mid \text{"+"} \mid \text{"\_"} \mid \text{"/"} \mid \text{"\$"} \mid \text{"\#"} \mid \text{":"} \mid \text{";"} \mid \text{"@"} \mid \text{"\%"} \mid \text{"0"}$$
$$\mid \text{string constants extracted from I/O examples}$$
$$\text{Integer constants } J := 0 \mid 1 \mid 2 \mid 3 \mid 99$$
$$\text{Input } X := x_1 \mid \ldots \mid x_k$$

Figure 6: The string manipulation DSL used in our experiments, taken from Odena et al. (2021).

We then compute the modulating effect of the relation upon the attention coefficient, $r_{ij}^l$, as

$$r_{ij}^l = R^l \, \text{EMBED}(\text{UNIQUEID}(p, i, j)) \tag{3}$$

where EMBED is a learned embedding of the sixteen possible values of UNIQUEID, and $R^l$ is a weight matrix that depends on the transformer layer $l$. We use four of these relational transformer layers with an embedding size of 512, hidden size of 2048, and 8 attention heads. After running the transformer we have a $10 \times 512$ tensor (number of entities by embedding dimension). We flatten this tensor and linearly project to a 512 tensor to produce the final embedding of the relation we are encoding.

## B  STRING MANIPULATION DSL

Our string manipulation DSL comes from BUSTLE (Odena et al., 2021) and is shown in Figure 6. It involves standard string manipulation operations, basic integer arithmetic, and if-then-else constructs with boolean conditionals. As an example, to compute a two-letter acronym, where we want to transform "product area" into "PA" and "Vice president" into "VP", we could use the program `Upper(Concatenate(Left(in0, 1), Mid(in0, Add(Find(" ", in0), 1), 1)))`.

BUSTLE extracts long repeatedly-appearing substrings of the I/O examples to use as constants, and we use this heuristic in CROSSBEAM as well for a fair comparison.

## C  EFFECT OF LARGER SEARCH BUDGET IN THE STRING DOMAIN

In our main string manipulation experiments (Section 4.1), we use a budget of 50,000 candidate programs to compare approaches. As shown in Figure 3, CROSSBEAM outperforms BUSTLE:

- BUSTLE solves 15 / 38 of their new tasks, and 44 / 89 of the SyGuS tasks, or 59 / 127 tasks total.
- CROSSBEAM solves 28.8 / 38 of their new tasks, and 66.6 / 89 of the SyGuS tasks, or 95.4 / 127 tasks total (averaged over 5 runs).

We repeat this experiment with a budget of 1 million candidates:

- BUSTLE solves 24 / 38 of their new tasks, and 70 / 89 of the SyGuS tasks, or 94 / 127 tasks total.
- CROSSBEAM solves 31.6 / 38 of their new tasks, and 73.0 / 89 of the SyGuS tasks, or 104.6 / 127 tasks total (averaged over 5 runs).

CROSSBEAM again outperforms BUSTLE. It is encouraging to see that CROSSBEAM continues to provide significant benefits for very difficult problems requiring a large amount of search.

Table 1: String manipulation results versus wallclock time, attempting to adjust time limits to account for different implementation languages.

|  | Time limit (s) | 38 New Tasks | 89 SyGuS Tasks | 127 Total Tasks |
|---|---|---|---|---|
| CROSSBEAM (Python) | 30 | 26.6 | 65.2 | 91.8 |
| Baseline enum (Python) | 30 | 20 | 54 | 74 |
| Baseline enum (Java) | 3.4 | 21 | 53 | 74 |
| BUSTLE (Java) | 3.4 | 29 | 72 | 101 |
| Baseline enum (Java) | 30 | 26 | 65 | 91 |
| BUSTLE (Java) | 30 | 32 | 80 | 112 |

## D  WALLCLOCK TIME COMPARISON IN THE STRING DOMAIN

Despite CROSSBEAM's impressive performance when controlling for the number of expressions considered, CROSSBEAM does not quite beat BUSTLE when we control for wallclock time. Note that CROSSBEAM is implemented in Python, while for BUSTLE we use the authors' Java implementation, which is faster in general than Python. To get a rough sense of the impact, we implemented the baseline bottom-up search in Python and found that it obtains very similar synthesis performance as the Java implementation from BUSTLE when controlling for the number of candidate programs considered, but the Python version is about $9\times$ slower than the Java one: our Python baseline solves 74 total tasks within 30 seconds, and the Java baseline solves 74 tasks within 3.4 seconds (but 73 in 3.3 seconds, and 75 in 3.5 seconds). This serves as a reference point for the difference in speed between the languages, but since no neural models are involved in the baselines, it is not directly applicable to the CROSSBEAM versus BUSTLE time comparison which does involve models. Nevertheless, Table 1 compares Python implementations with a 30 second time limit versus Java implementations with a 3.4 second time limit.

Even when we do our best to adjust the time limits fairly, BUSTLE still solves more tasks than CROSSBEAM. However, the original BUSTLE implementation has other optimizations: the BUSTLE model has a simple and small architecture, the method was designed to enable batching of the model predictions, operations in the string manipulation domain are extremely quick to execute, and the entire system is implemented in Java. In contrast, CROSSBEAM is implemented in Python, and our work as a whole is more focused on searching efficiently in terms of candidate programs explored instead of wallclock time, favoring improvements in deep learning methodology and ease of iteration on modeling ideas over runtime speed. In CROSSBEAM, batching model predictions for all operations in an iteration and batching the UniqueRandomizer sampling are both feasible but challenging engineering hurdles that would lead to speedups, but are currently not implemented.

## E  ADDITIONAL INDUCTIVE LOGIC PROGRAMMING RESULTS

For the ILP experiments, we compare CROSSBEAM, the baseline bottom-up enumerative search, two variations of Metagol (using metarules corresponding to our logic programming DSL, or using the 6 metarules suggested in the official repo's README), and two variations of Popper (with and without predicate invention (Cropper & Morel, 2021b)).

To test the ability of our model to learn arithmetic relations, we handcrafted a small corpus of 30 arithmetic problems. These problems were removed from the training data. For all of the methods being compared, we measure whether a program implementing the target relation can be found within a 30 second time limit. The results are shown in Figure 7, and we see that CROSSBEAM solves the most tasks among all methods considered.

Additionally, on the randomly-generated test set, we observed that CROSSBEAM usually solves problems incredibly quickly (in under 2 seconds) or not at all within 30 seconds, while the other approaches, especially the baseline enumerative search, can take much longer to find solutions. This is illustrated in Figure 8, which shows the gap between CROSSBEAM and other methods increasing as the time limit decreases from 30 seconds to 1 second.

| Relation | CROSSBEAM | Enum | Metagol[1] | Metagol[2] | Popper[1] | PopperInv[1] | Popper[2] | PopperInv[2] |
|---|---|---|---|---|---|---|---|---|
| $|x-y|=1$ | ✔ | ✔ | ✔ | ✔ | ✔ | ✔ | ✔ | ✔ |
| $|x-y|=2$ | ✔ | ✔ | ✔ | ✔ | ✔ | ✔ | ✔ | ✔ |
| $|x-y|=3$ | ✔ | ✗ | ✔ | ✗ | ✔ | ✔ | ✗ | ✗ |
| $|x-y|=4$ | ✔ | ✗ | ✔ | ✗ | ✗ | ✗ | ✗ | ✗ |
| $|x-y|=5$ | ✔ | ✗ | ✗ | ✗ | ✗ | ✗ | ✗ | ✗ |
| $|x-y|=6$ | ✗ | ✗ | ✗ | ✗ | ✗ | ✗ | ✗ | ✗ |
| $|x-y|=7$ | ✗ | ✗ | ✗ | ✗ | ✗ | ✗ | ✗ | ✗ |
| $x \bmod 2 = 0$ | ✔ | ✔ | ✗ | ✗ | ✔ | ✔ | ✔ | ✔ |
| $x \bmod 3 = 0$ | ✔ | ✔ | ✗ | ✗ | ✔ | ✔ | ✔ | ✔ |
| $x \bmod 4 = 0$ | ✔ | ✔ | ✗ | ✗ | ✗ | ✔ | ✗ | ✗ |
| $x+2=y$ | ✔ | ✔ | ✔ | ✔ | ✔ | ✔ | ✔ | ✔ |
| $x+3=y$ | ✔ | ✔ | ✔ | ✔ | ✔ | ✔ | ✔ | ✔ |
| $x+4=y$ | ✔ | ✔ | ✔ | ✔ | ✗ | ✔ | ✗ | ✔ |
| $x+5=y$ | ✔ | ✔ | ✔ | ✗ | ✗ | ✔ | ✗ | ✗ |
| $x+6=y$ | ✔ | ✔ | ✔ | ✗ | ✗ | ✔ | ✗ | ✔ |
| $x-2=y$ | ✔ | ✔ | ✔ | ✔ | ✔ | ✔ | ✔ | ✔ |
| $x-3=y$ | ✔ | ✔ | ✔ | ✔ | ✔ | ✔ | ✔ | ✔ |
| $x-4=y$ | ✔ | ✔ | ✔ | ✔ | ✗ | ✔ | ✗ | ✔ |
| $x-5=y$ | ✔ | ✔ | ✔ | ✗ | ✗ | ✔ | ✗ | ✗ |
| $x-6=y$ | ✔ | ✔ | ✔ | ✗ | ✗ | ✔ | ✗ | ✗ |
| $x/2=y$ | ✔ | ✔ | ✗ | ✗ | ✗ | ✗ | ✗ | ✗ |
| $x/3=y$ | ✔ | ✔ | ✗ | ✗ | ✗ | ✗ | ✗ | ✗ |
| $x/4=y$ | ✔ | ✗ | ✗ | ✗ | ✗ | ✗ | ✗ | ✗ |
| $x*2=y$ | ✔ | ✔ | ✗ | ✗ | ✗ | ✗ | ✗ | ✗ |
| $x*3=y$ | ✔ | ✔ | ✗ | ✗ | ✗ | ✗ | ✗ | ✗ |
| $x*4=y$ | ✔ | ✗ | ✗ | ✗ | ✗ | ✗ | ✗ | ✗ |
| $x<y$ | ✔ | ✔ | ✔ | ✔ | ✔ | ✔ | ✔ | ✔ |
| $x>y$ | ✔ | ✔ | ✔ | ✔ | ✔ | ✔ | ✔ | ✔ |
| $x \leq y$ | ✔ | ✔ | ✔ | ✔ | ✗ | ✗ | ✔ | ✔ |
| $x \geq y$ | ✔ | ✔ | ✔ | ✔ | ✗ | ✗ | ✔ | ✔ |
| Total Solved | **28** | 23 | 18 | 12 | 11 | 18 | 12 | 15 |

Figure 7: Testing on 30 unseen handcrafted test problems. We report whether each system can synthesize a logic program implementing the target relation, given a 30 second time limit. CROSSBEAM uses the Transformer architecture from Appendix A.2 (the MLP architecture solves 1 fewer task). Metagol[1] uses metarules corresponding to our logic programming DSL, while Metagol[2] uses the 6 metarules suggested in the official repo's README. PopperInv uses predicate invention while Popper does not. Popper[2] and PopperInv[2] have access to the same primitive predicates that CROSSBEAM does (iszero, successor, and isequal) while Popper[1] and PopperInv[1] only have access to iszero and successor. In principle, logic programs can implement equality by reuse of variables, so we ran both with and without the isequal predicate.

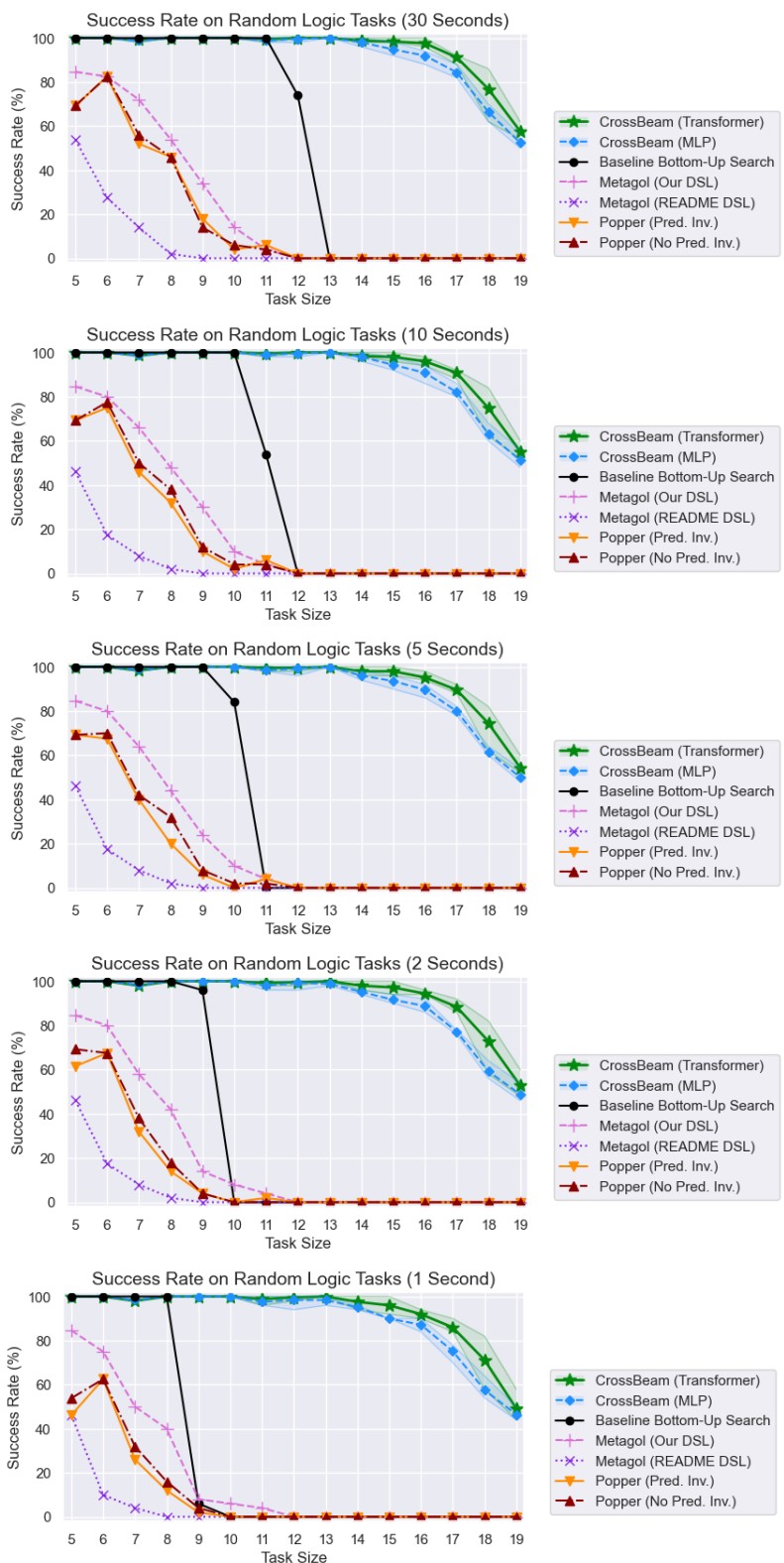

Figure 8: Results on the randomly-generated logic tasks, for different time limits. As the time limit gets shorter, the gap between CROSSBEAM and other methods increases.

