# OpenReview forum: "CrossBeam: Learning to Search in Bottom-Up Program Synthesis"
_ICLR.cc/2022/Conference — ICLR 2022 Poster_

### Official Review · Reviewer_D2Hc · 2021-11-01

**Correctness:** 3
**Technical Novelty And Significance:** 3
**Empirical Novelty And Significance:** 3
**Recommendation:** 8
**Confidence:** 5

**Main Review:**

This paper makes a step in the right direction to advance the state of the art in learning systems for solving program synthesis problems. Namely, CrossBeam leverages the search context to decide which program to evaluate next in search. Previous methods used functions for guiding the bottom-up synthesis (e.g., Bustle and Probe) that accounted only for a program in isolation (i.e., is this program likely to used as part of a solution?).

CrossBeam has two major weaknesses, however. One of the weaknesses is fully discussed in the paper, while the other is only partially discussed.

The first weakness of CrossBeam is the system's running time. The paper does a good job explaining that CrossBeam is much slower than Bustle, a bottom-up search competitor that uses a much simpler model to guide the search. If CrossBeam and Bustle are compared in terms of programs evaluated, then CrossBeam is more effective. If the systems are evaluated in terms of running time, then Bustle is more effective. I understand the authors argument that they are seeking to advance our understanding of how neural models can be used to aid the synthesis process. I agree with the authors and also see value in the kind of work that is presented in this paper.

Other research areas such as Classical Planning have dealt with similar issue for many years. The basic evaluation metric in that community is running time. The solution that that community found was to accept papers that present novel ideas and that do not necessarily advance the state of the art. I think this is what we should do here, accept this paper for presenting an interesting idea. I would have liked the paper better if it was written with the angle of presenting a novel idea that doesn't necessarily advance the state of the art.

This discussion leads me to the second weakness of CrossBeam. The search algorithm CrossBeam uses isn't ideal because it is incomplete and its search can stall. The paper has an interesting discussion around this issue, on how Beam Search as a decoding system for CrossBeam's model can cause the search to stall. The solution used in the paper is the use UniqueRandomizer by Shi et al. to attempt to see different programs as the search progresses. CrossBeam with the UniqueRandomizer indeed performs better than CrossBeam with Beam Search. However, what isn't clear and isn't discussed in the paper is whether the UniqueRandomizer solves the stalling issue CrossBeam faces.

The lack of information on whether UniqueRandomizer solves the stalling issue is troublesome because the experiments on the string synthesis problems use a suspiciously low search budget: 50,000 programs. Such a small search budget allows for the solution of only the easier problems from the problem set. What happens when we increase the search budget to millions, so we can also solve the harder instances in the set? Does CrossBeam's search eventually stall even with the UniqueRandomizer method?

My last concern is that it might be hard to reproduce the results presented in the paper. It isn't clear if the source code will eventually be made available to the research community. When it comes to reproducing the results of the paper, the paper misses important information. For example, it isn't clear how the embedding z_i is defined.

**Summary Of The Paper:**

This paper introduces CrossBeam, a method for guiding the synthesis of bottom-up search for solving program synthesis problems. The main novelty of CrossBeam is a neural architecture that leverages the search context to decide which program to evaluate next in the bottom-up search. Empirical results in string synthesis problems and ILP show that CrossBeam is superior to other search methods in terms of number of programs evaluated.

**Summary Of The Review:**

The paper presents an interesting neural architecture that allows the search to account for the search state while deciding which program to evaluate next. The paper has two main weaknesses: the proposed method is much slower than other methods from the literature and the search algorithm can stall during search and not make progress, despite being allowed more computational time. The experiments possibly miss the fact that CrossBeam might suffer from getting stuck even when using the UniqueRandomizer. This is because the search budget allowed to the systems is suspiciously small.

---

> ### Author Response · Authors · 2021-11-19
> **Author response to Reviewer D2Hc**
>
> Thank you for your review!
>
> > The search algorithm CrossBeam uses isn't ideal because it is incomplete and its search can stall. ... However, what isn't clear and isn't discussed in the paper is whether the UniqueRandomizer solves the stalling issue CrossBeam faces.
>
> CrossBeam with beam search can stall by entering an infinite loop where no progress can be made. Sampling argument lists, for example with UniqueRandomizer, does solve the stalling problem, because there is always some probability of choosing an argument list that leads to progress. However, if the model predicts poorly for the task, it may take a long time (many samples) for this progress to occur.
>
> > What happens when we increase the search budget to millions, so we can also solve the harder instances in the set? Does CrossBeam's search eventually stall even with the UniqueRandomizer method?
>
> Yes, CrossBeam with UniqueRandomizer continues to solve more problems as we increase the search budget to 1 million candidates explored, reaching an average (over 5 runs) of 9.2 more problems solved total over both sets of benchmarks compared to the budget of 50,000 candidates in the paper.
>
> > My last concern is that it might be hard to reproduce the results presented in the paper. It isn't clear if the source code will eventually be made available to the research community.
>
> We will open-source the code and trained models if the paper is accepted.
>
> > it isn't clear how the embedding z_i is defined
>
> We use an embedding lookup table mapping the value V_i’s integer size to the trainable embedding z_i. This is explained at the end of the “Value module” paragraph in Section 3.1.

---

> > ### Comment · Reviewer_D2Hc · 2021-11-22
> > **How Does CrossBeam Compare With Bustle in the Long Run?**
> >
> > Thanks for your reply!
> >
> > >Yes, CrossBeam with UniqueRandomizer continues to solve more problems as we increase the search budget to 1 million candidates explored, reaching an average (over 5 runs) of 9.2 more problems solved total over both sets of benchmarks compared to the budget of 50,000 candidates in the paper.
> >
> > And how many problems can Bustle solve with a budget of 1 million candidates?

---

> > > ### Author Response · Authors · 2021-11-23
> > > **Number of problems solved with a budget of 1 million candidates**
> > >
> > > With a budget of 50,000 candidates:
> > > * BUSTLE solves 15 / 38 of their new tasks, and 44 / 89 of the SyGuS tasks, or 59 / 127 tasks total.
> > > * CrossBeam solves 28.8 / 38 of their new tasks, and 66.6 / 89 of the SyGuS tasks, or 95.4 / 127 tasks total (averaged over 5 runs).
> > >
> > > With a budget of 1 million candidates:
> > > * BUSTLE solves 24 / 38 of their new tasks, and 70 / 89 of the SyGuS tasks, or 94 / 127 tasks total.
> > > * CrossBeam solves 31.6 / 38 of their new tasks, and 73.0 / 89 of the SyGuS tasks, or 104.6 / 127 tasks total (averaged over 5 runs).
> > >
> > > Thus, CrossBeam outperforms BUSTLE in both cases.

---

### Official Review · Reviewer_cqKb · 2021-11-02

**Correctness:** 3
**Technical Novelty And Significance:** 3
**Empirical Novelty And Significance:** 3
**Recommendation:** 6
**Confidence:** 3

**Main Review:**

Overcoming exponential search complexity in program synthesis is an important problem. Reducing search complexity could significantly improve synthesis accuracy and enable practical applications. The paper is well-written and relatively easy to follow.

Taking into account the full search history can potential be bottleneck if applied to general-purpose programming languages or DSLs with larger number of ops. Intuitively, not all previously searched programs would make good candidates. Have you considered pruning away previously searched programs that could not make a subprogram (perhaps a non-subexpressions)? Have you analyzed computational complexity of CrossBeam as a function of number of ops in a DSL, the number of I/O examples? This could help understand if including the full search history as part of context can lead to bottlenecks of that kind.

In the value module, what are the typical values of |S|? How does it scale with number of ops in a DSL?

I believe the search history can be maintained with left-to-right models. It would be interesting to see this as an ablation study. One can think of a beam search algorithm with backtracking where for a target expression f(A, B) the search could recover from mistakes in predicting A, and continue to search the optimal solution. This could enable applications of variants of CrossBeam for language models like Codex.

What is the loss function utilized during training?

Given the correct ground truth program can be reached by combining previously explored programs in various orders, this might have a significant impact on the runtime. Can a penalty be introduced as part of loss function to favor smaller depth of search?

It would be interesting to compare CrossBeam beam-aware training to an RL based on-policy training method like e.g. proximal policy optimization (PPO) method. RL allows to construct a reward function to optimize for a specific target evaluation metric. Because indeed, even though it uses samples generated by model the optimization objective is not that used during testing. An RL approach could further reduce discrepancy between training and test setting although not absolutely necessary.

Is teacher forcing used during model training?

Wallclock time comparison does not makes sense if you compare implementations in different programming languages (Java and Python). The differences in execution speeds maybe significant. It is better to compare FLOPs or a metric independent on programming language.

Minor:
"autoregressive model like an LSTM" - an LSTM model can follow an autoregressive formulation/be trained autoregressively, but does not have to be in general case.

Figure2 is confusing and needs more explanation.

**Summary Of The Paper:**

The paper introduces CrossBeam – a variant of bottom-up enumerative search algorithm for program synthesis guided by a pointer network. The neural network is trained on-policy, utilizing search histories and intermediate program executions as context. The neural network aims to help choose how to combine previously explored subprograms and subsequently reduce the search complexity. Specifically, given a DSL op, I/O examples, and search histories the neural network models a distribution of argument lists from which K most likely argument lists are sampled during CrossBeam search, and an op is executed with these arguments list to verify correctness on I/O examples.

The method is evaluated on the program synthesis task (programming by example setting) in two different domains: string manipulation and logic programming. CrossBeam significantly reduces program search space compared to state of the art, and improves accuracy compared to previous art.



**Summary Of The Review:**

An interesting and well written paper that suggests a solution to a long standing problem in program synthesis from input/output examples. Some machine learning design decisions appear overly complex, although motivated. It would strengthen the paper if authors could rethink the ML design having general purpose programming languages in mind or different DSLs (would including an entire search history as context still scale?). An ablation study connecting this approach to the left-to-right models (e.g. use beam search with backtracking) could give ideas about feasibility of practical applications of such models.

---

> ### Author Response · Authors · 2021-11-19
> **Author response to Reviewer cqKb (part 1)**
>
> Thank you for your review!
>
>
> > Have you considered pruning away previously searched programs that could not make a subprogram (perhaps a non-subexpressions)?
>
> We have actually considered the idea of thinning down the set of explored values, possibly using a BUSTLE-like model that predicts whether a value will be used in a solution, and then “forgetting” values we have explored but are predicted to be not useful. This would be similar to the “garbage collection” idea from [1]. However, considering that the set of explored values is essentially a vocabulary for the CrossBeam sequence model, but pruning the vocabulary of sequence models (e.g., language or translation models) is not standard practice to our knowledge, we chose not to explore this direction for now.
>
> [1] Amit Zohar and Lior Wolf. Automatic program synthesis of long programs with a learned garbage collector. NeurIPS 2018.
>
> > Have you analyzed computational complexity of CrossBeam as a function of number of ops in a DSL, the number of I/O examples?
>
> The CrossBeam search scales linearly with the number of I/O examples, since during search we execute operations for all I/O examples. Each iteration of the outermost loop (line 3 in Algorithm 1) scales linearly in the number of DSL operations (since we iterate over all operations). However, it is not easy to analyze how many outermost loop iterations are required to solve a problem, since this depends primarily on how difficult the problem is to solve (how large is the minimal solution) and how good the model predictions are.
>
> > In the value module, what are the typical values of |S|? How does it scale with number of ops in a DSL?
>
> |S| is the number of candidates explored at that time. In our experiments we bound |S| by 50,000 and say that a method fails to solve a task if a solution is not found by then. Figure 3 shows how often benchmark tasks are solved within some number of candidates explored. The number of candidates considered by one iteration of the outermost loop (line 3 in Algorithm 1) scales linearly in the number of DSL operations (since we iterate over all operations).
>
> > I believe the search history can be maintained with left-to-right models. It would be interesting to see this as an ablation study. One can think of a beam search algorithm with backtracking where for a target expression f(A, B) the search could recover from mistakes in predicting A, and continue to search the optimal solution. This could enable applications of variants of CrossBeam for language models like Codex.
>
> This is an interesting idea. We think it is nontrivial to implement a CrossBeam-like approach for a language model like Codex, but it is an exciting possibility for future work.
>
> > What is the loss function utilized during training?
>
> The loss is the negative log likelihood of the ground-truth argument lists in the training data. We added a formula to Section 3.2 for the loss function in our revision.
>
> > Given the correct ground truth program can be reached by combining previously explored programs in various orders, this might have a significant impact on the runtime. Can a penalty be introduced as part of loss function to favor smaller depth of search?
>
> The model already tries to construct the program in as few iterations as possible. Furthermore, the model is aware of the complexity for creating a particular value, since a value’s size (number of expression tree nodes) is part of its embedding. Hence, the model is doing its best to use the search context to reach a solution as quickly as it can.
>
> > It would be interesting to compare CrossBeam beam-aware training to an RL based on-policy training method like e.g. proximal policy optimization (PPO) method. ... An RL approach could further reduce discrepancy between training and test setting although not absolutely necessary.
>
> Methods from reinforcement learning would seem appropriate, given that search is essentially a sequential decision process. We actually experimented with using DAGGER [2] to train the system. Specifically during our on-policy training data collection, we would have a certain chance to not add the ground-truth trace element to the set of explored values, but rather rely only on the current model to expand the search history. When setting this chance to be 20%, we found that the model trains 2x slower (more search context is collected and could be larger), but it did not sufficiently improve the performance on our synthetic validation set. Our hunch is that the hands-on supervision from beam-aware training helps bootstrap learning, but it’s possible that we could fine-tune our model with PPO, DAGGER or other RL methods with a curriculum, and that this would help address the train/test discrepancy.
>
> [2] Ross et.al, A Reduction of Imitation Learning and Structured Prediction to No-Regret Online Learning, AISTATS 2011

---

> ### Author Response · Authors · 2021-11-19
> **Author response to Reviewer cqKb (part 2)**
>
> > Is teacher forcing used during model training?
>
> We would say no, because of the way that search is run during training. If the model makes a mistake at some point in training, then the incorrect prediction is added to the set of available values, so that in future iterations, the model is trained to learn not to expand the incorrect value into larger expressions. This is very different from what happens during teacher forcing in the autoregressive language modeling sense. To clarify this, we have added an equation for the loss function in 3.2; the denominator of p(arglist | ... ) in that equation involves summing over all the values in S, which include the incorrect values predicted by the model.
>
> > Wallclock time comparison does not makes sense if you compare implementations in different programming languages (Java and Python). The differences in execution speeds maybe significant. It is better to compare FLOPs or a metric independent on programming language.
>
> It is difficult to compare directly using FLOPs because the search algorithms are quite different and use different amounts of CPU compute.
>
> > Minor: "autoregressive model like an LSTM" - an LSTM model can follow an autoregressive formulation/be trained autoregressively, but does not have to be in general case.
>
> We have revised to avoid implying that LSTMs are always autoregressive.
>
> > Figure2 is confusing and needs more explanation.
>
> We have revised Figure 2 to delineate the search context summary module and the argument selector module, so that all 4 modules described in Section 3.1 are labeled in the figure.

---

### Official Review · Reviewer_tHmt · 2021-11-03

**Correctness:** 3
**Technical Novelty And Significance:** 4
**Empirical Novelty And Significance:** 3
**Recommendation:** 8
**Confidence:** 4

**Main Review:**

Strengths:
- A novel and interesting approach to neural program synthesis, based on a combination of existing techniques from the literature such as bottom-up program synthesis, on-policy beam aware training, UniqueRandomizer, and pointer networks.
- Thorough experimental analysis shows very promising results across two important benchmarks from the literature (string manipulation and inductive logic programming). The ablation study help demonstrate the importance of the different components.

Weaknesses (Minor):
- The wall clock comparison does not exhibit similar improvement over baseline (BUSTLE) as the number of expressions due to lack of batching. While the authors claim batching is feasible and would lead to speed up, there is no experimental evidence of that.
- Some details about the experiments are not entirely clear (see questions below).


Question:
1) Comparison to RobustFill: It is not clear to me how are the results compared? What does it mean that RobustFill considered 50,000 candidate programs? Are these generated them using a beam search with a beam width of 50,000?
2) Random training ablation: "obtain argument lists by randomly sampling values from the set S of explored values" - sample uniformly?
3) What is in the set Consts?
4) Current approach iterates over all operators in each iteration. Did you consider also learning to select/prioritize operators based on the specification and the pool of existing subprograms?


**Summary Of The Paper:**

The paper presents crossbeam, a bottom-up program synthesis with neural-guided search. For each of the supported operators in the DSL, a deep neural network is trained to predict which subset of previously computed subprograms (argument lists) to combine using the operator. Using UniqueRandomizer (that allows sampling sequences incrementally and without replacement), a K new unique subprograms are generated and added to the pool of subprograms that can later be combined again until a solution that satisfy the input-output specification is found. Training is done on-policy using beam-aware training.

**Summary Of The Review:**

Overall, I think the paper presents a novel and interesting approach that shows significant improvement, however there are still engineering challenges related to batching.

---

> ### Author Response · Authors · 2021-11-19
> **Author response to Reviewer tHmt**
>
> Thank you for your review!
>
> > Comparison to RobustFill: It is not clear to me how are the results compared? What does it mean that RobustFill considered 50,000 candidate programs? Are these generated them using a beam search with a beam width of 50,000?
>
> Yes. These results are borrowed from the BUSTLE paper, which used a single beam search to gather all of RobustFill’s candidate programs for a given task.
>
> > Random training ablation: "obtain argument lists by randomly sampling values from the set S of explored values" - sample uniformly?
>
> Yes.
>
> > What is in the set Consts?
>
>
> For the string manipulation domain, we use the same constants as in BUSTLE. This includes common symbol or delimiter characters that appear at least once in the I/O example, long substrings that appear multiple times in the I/O example, and a few hardcoded integers. Please see Appendix B (in particular Figure 6) for more details.
>
> For the logic domain, the constants are the zero, succ, and eq relations listed in Figure 4A.
>
> More broadly, constants are any values considered by the search that appear at expression tree leaves but are not input variables. In the general formulation, these constants are given by the DSL, as described in Section 2 and the Input section of Algorithm 1 (“DSL L describing constants Consts and operations Ops”).
>
> > Current approach iterates over all operators in each iteration. Did you consider also learning to select/prioritize operators based on the specification and the pool of existing subprograms?
>
> We considered a variation of the model architecture where the model predicted the operation as part of its output sequence. However, in preliminary experiments this approach did not perform as well, about 12% worse on the synthetic validation set compared to iterating over all the operators. We believe it is harder for the model to perform well in this formulation -- at every step of the search, the model must accurately predict which operations to use given only the search context, since any operations that do not appear in the beam are not searched at all. We believe that by iterating over all operators, we add a bit more robustness to the CrossBeam search.

---

> > ### Comment · Reviewer_tHmt · 2021-11-30
> > **Thank you for the response**
> >
> > Thank you for the response and for clarifying all the details!

---

### Official Review · Reviewer_Q8M3 · 2021-11-04

**Correctness:** 4
**Technical Novelty And Significance:** 3
**Empirical Novelty And Significance:** Not applicable
**Recommendation:** 8
**Confidence:** 4

**Main Review:**

Strengths
- The empirical results are quite strong and show clear improvements on prior work. There is a blemish in that the wall-clock performance is not as good compared to BUSTLE, so it is unclear whether the proposed method will actually end up being more practically useful. Nevertheless it doesn't seem unreasonable to compare on the basis of the number of candidate programs.
- The paper contains useful ablation experiments that control for various aspects of the method.
- The presentation is very clear. The paper contains easy to understand diagrams and Algorithm 1 was particularly helpful for understanding the training and inference procedure (and how they relate to each other).

Weaknesses
- A more comprehensive presentation of related methods would have been helpful, for example in a tabular format. The current presentation in Section 5 mostly emphasizes one or two differences of this work compared to various previous works. I believe it would be more useful to readers if the section can explain how all the works relate to each other.
- Unlike other neural program synthesis methods, in particular RobustFill, the method cannot handle any mistakes made in the input-output specification.

Questions
- In general, there will be many programs that can satisfy the input-output examples, but only some of them may accurately reflect the user's intent. How do we expect the proposed method to be able to find such programs, either based on some prior (simpler programs are better) or some inferences possible from the examples chosen by the user (e.g. similar to pragmatics in human communication)?


**Summary Of The Paper:**

The paper proposes a method for program synthesis from input-output examples using a bottom-up search method. Unlike previous works based on bottom-up enumerative search, which follow a strict ordering for exploring expressions based on their size, and eventually enumerates all possible programs up to a given size, this paper learns an entirely neural policy for the search procedure. This policy uses all of the available information at each step of the search, including the sub-programs identified so far, to decide which sub-program to explore next. It is trained using on-policy learning based on action traces created from randomly generated programs and inputs. The authors evaluate the method on string manipulation and  inductive logic programming benchmarks, and show strong empirical results compared to prior work.

**Summary Of The Review:**

I vote to accept the paper considering the strong empirical results and the interesting method.

---

> ### Author Response · Authors · 2021-11-19
> **Author response to Reviewer Q8M3**
>
> Thank you for your review!
>
> > Unlike other neural program synthesis methods, in particular RobustFill, the method cannot handle any mistakes made in the input-output specification.
>
> This is correct. However one could imagine extending the bottom-up search to output “close solutions” whenever it encounters a value that is very similar to the example outputs, indicating that the specified output might have a typo. In such an approach, one might also consider adding random errors to the example outputs for the synthetically-generated training tasks. While interesting, these ideas are orthogonal to the paper’s contribution.
>
> > In general, there will be many programs that can satisfy the input-output examples, but only some of them may accurately reflect the user's intent. How do we expect the proposed method to be able to find such programs, either based on some prior (simpler programs are better) or some inferences possible from the examples chosen by the user (e.g. similar to pragmatics in human communication)?
>
> Indeed, issues of inductive bias are important for learning programs from examples. Existing methods, based on learning a ranking/prior or the RSA model of pragmatic communication are interesting and important. Such approaches could compose with our work by reranking programs found via search--but search has to be tractable in the first place in order for such methods to come into play.

---

### Author Response · Authors · 2021-11-19
**Updated Paper Revision**

We thank the reviewers for their in-depth and helpful reviews. We have responded to each reviewer's comments. In our updated paper revision, we have added some clarifications including a formula for our loss function (Equation 1 in Section 3.2) and clearly delineating the 4 model modules in Figure 2.

---

### Decision · Program_Chairs · 2022-01-20

**Decision:**

Accept (Poster)

**Comment:**

This paper addresses the problem of program synthesis given input/output examples and a domain-specific language using a bottom-up approach. The paper proposes the use of a neural architecture that exploits the search context (all the programs considered so far and their execution results) to decide which program to evaluate next. The model is trained on-policy using beam-aware training and the method is evaluated on string manipulation and inductive logic programming benchmarks. The results show that the proposed method outperforms previous work in terms of the number of programs evaluated and accuracy.

Overall, the reviewers found the paper to be well-written and the idea proposed to be significantly novel and interesting to be presented at the conference and I agree. Several limitations were pointed out by the reviewers in terms of (i) actual run-time performance, (ii) the incompleteness of the search algorithm and the (iii) reproducibility of the approach. I believe the authors have addressed these points satisfactorily in their comments.